# Genome Size Variation Assessment in *Vitis vinifera* L. Landraces in Ibiza and Formentera (Balearic Islands)

**DOI:** 10.3390/plants11141892

**Published:** 2022-07-21

**Authors:** Raquel González, Joan Vallès, Teresa Garnatje

**Affiliations:** 1Laboratori de Botànica, Unitat Associada al CSIC, Facultat de Farmàcia i Ciències de l’Alimentació, Institut de la Biodiversitat IRBio, Universitat de Barcelona, Av. Joan XXIII 27-31, 08028 Barcelona, Catalonia, Spain; joanvalles@ub.edu; 2Institut Botànic de Barcelona (IBB), CSIC-Ajuntament de Barcelona, Passeig del Migdia s.n., Parc de Montjuïc, 08038 Barcelona, Catalonia, Spain; tgarnatje@ibb.csic.es

**Keywords:** Balearic Islands, cultivars, DNA amount, flow cytometry, Formentera, genome size, Ibiza, landraces, local varieties, *Vitis vinifera*

## Abstract

Plant genome size has many applications in different biological fields including ecology and plant breeding. The 2C value for *Vitis vinifera* L. has not been widely studied; furthermore, to date, no data from local landraces in the Pityusic Islands (the two smaller inhabited Balearic Islands, Ibiza, and Formentera) have been reported. This research aims to contribute to this knowledge and investigate whether there are variations between different grape landraces cultivated in Ibiza and Formentera and also among the same landraces on each island. To this end, 36 accessions of 15 cultivars and 6 landraces, identified with SSR markers, were assessed using flow cytometry. The results revealed that 2C values ranged from 1.09 pg to 1.28 pg. There were statistically significant differences in ‘AG1’ and ‘AG2’ landraces and ‘Santa Magdalena’, ‘Garnatxa’, ‘Danugue’, and ‘Valencí tinto/Grumier’ cultivars. No statistically significant differences were found in terms of the genome size content between islands. Statistically significant differences were found in accessions within ‘AG2’ landrace group and ‘Beba’ cultivar. The results presented here constitute the first-ever reported information on genome size in the genus *Vitis vinifera* in Pityusic, Balearic, and, in general, Spanish accessions, and they are one of the largest prospections in this field for this species anywhere. Further research should be conducted to explain the differences in nuclear DNA content found between landraces and cultivars studied here with others cultivated in different islands or countries to understand whether genome size varies in modern cultivars compared with local landraces. Additionally, it would be interesting to investigate whether there is a relationship between genome size and adaptations to diverse climatology conditions, crop management, and ripening characteristics.

## 1. Introduction

*Vitis vinifera* L. (Vitaceae) is one of the most important cultivated plants in socio-economic terms. In its annual report, the International Organisation of Vine and Wine [1] revealed that the global surface area of this crop is over 7.4 mha. Spain is the first country in the world, with the largest surface area dedicated to grape cultivation, 964 kha [1], and together with four other countries, namely France, China, Italy, and Turkey, represents half of the vineyards of the earth [1]. In the Balearic Islands, the area coverage of vineyards amounts to 2336 ha [2]. Ibiza and Formentera, which are the two smaller inhabited Balearic Islands, separated from each other by only 19 km and also known as Pityusic Islands, contribute to a small part of this surface, with 58.6 ha in Ibiza Island (also known by its Catalan name Eivissa) [3] and 14.3 ha in Formentera Island [4].

Despite the importance of this crop around the world, *Vitis vinifera* biodiversity represented in them is scarce. Only 35 cultivars are used in 66% of the world’s vineyards when there are more than 2000 cultivars available [5]. In Spain, this variability is even more reduced, with no more than 10 cultivars covering 74.3% of the total vineyard surface [1]. The Balearic Islands are not an exception to this pattern: ‘Cabernet’, ‘Sauvignon’, ‘Chardonnay blanc’, ‘Merlot’, and ‘Syrah’ account for 39% of Balearic Islands’ vineyards [6]. Although there is 38% of the vineyards with local landraces such as ‘Manto negro’, ‘Callet’, and ‘Pensal Blanca’, the remaining 23% consist of other varieties not specified. In Ibiza, the main cultivars are ‘Monestrell’ and ‘Garrut’, with 76.45% of the crops, and in Formentera, 57.5% consist of ‘Monestrell’, ‘Garrut’, and ‘Garnatxa tinta’. There are still 7.98% unknown varieties in Ibiza, and 6% unknown varieties in Formentera [7]. This low variability can be explained by the fact that Spanish legislation does not allow farmers to grow varieties other than the ones registered in the Spanish Commercial Variety or other official European databases [8].

Despite this reduced biodiversity, it should be borne in mind that the vine is a typical Mediterranean crop, and therefore, it was and still is present in almost every farmhouse on the Balearic Islands. Pliny, in his *Natural History*, was the first to mention the exquisiteness of Ibizan’s wines, and all along history, wines and grapes from the Balearic Islands have been famous [9].

In the last 20 years in the Balearic Islands, in order to preserve biodiversity and add variability to the local winemaking industry, there has been a great effort from Institut de Recerca i Formació Agrària i Pesquera (IRFAP) and Balearic Islands University (UIB) on prospecting and studying grape biodiversity in the islands [10,11,12]. SSR markers have been used in several of these studies [6,13,14,15] to identify grape diversity, as in some cases some vines have local names, but, once analysed with SSR markers, they are found to be registered cultivars and not autochthonous landraces as it had been presumed. Still, Ibiza and Formentera are scarcely represented in those studies, and in none of them, genome size has been evaluated.

Genome size, the nuclear DNA amount, was first described by Swift [16] as the C value, i.e., the DNA content of an unreplicated gametic nucleus or chromosome set. In this term, ‘C’ accounts for ‘constant’, since it was assumed that it was a species-specific characteristic. Nevertheless, even if this parameter continues to be a good characteristic trait for many taxa, a certain level of genuine, non-artefactual intraspecific variation occurs, importantly due to chromosomal rearrangements and polyploidy [17,18]. Genome size variation is ca. 2400-fold in embryophytes or land plants, although it is still understudied since, for instance, it is only known for ca. 3% of the angiosperms, which are, with more than 350,000 species recorded to date, the largest group of land plants [18]. Apart from its knowledge being fundamental in order to undertake DNA sequencing projects, nuclear DNA content and its possible variation are interesting, as it is correlated with a number of abiotic and biotic traits and has many applications in different fields such as ecology, evolution, conservation, or plant breeding [19,20,21,22]. Only six works assessing genome size have been conducted in *Vitis vinifera* since the early 1990s [21,23,24,25,26,27]. Four of them exclusively studied *Vitis* species, while the other two assessed the DNA amount in plants from other genera as well. None of them evaluated genome size in local vine landraces from the Balearic Islands, and neither used as many accessions as the present study.

We undertook genome size analysis on 36 accessions of 15 cultivars and 6 landraces to characterise the Pityusic landraces from this viewpoint so that they could be compared with other cultivars and races worldwide and to detect and evaluate possible (1) intraspecific variations and (2) variations linked to distribution.

## 2. Results and Discussion

Data on nuclear DNA content for each accession are presented in Table 1. The results presented here constitute the first-ever reported information on genome size in the genus *Vitis* in Pityusic, Balearic, and, in general, Spanish accessions. The 2C values ranged from 1.090 pg for ‘Garnatxa’ to 1.277 pg for ‘Santa Magdalena’, which means that the 1C genome size ranged from 533.01 to 624.453 Mbp (see Table 1 for the detailed values for each accession). The whole-genome sequence of *Vitis vinifera* L. has been estimated at around 500 Mbp [21,24,26]. Our results were not far off, considering that the estimation methods were different and that we prospected not one but several accessions from different cultivars and landraces. As previously mentioned (Section 2.2), a previous paper dealing with genome size assessment in *Vitis vinifera* using flow cytometry also showed variation and did not largely differ from the range presented here. The assessments were of good quality, as indicated by a mean HPVC value of 4.17%.

### 2.1. Comparison with Previous Results on Vitis Genome Size

Compared with the results of Arumuganathan and Earle [23] and Lodhi and Reisch [24] regarding the 2C content for *Vitis vinifera*, which ranged from 0.92 pg to 1.00 pg, our results were higher in all the studied accessions. When compared with the results reported in Leal et al. [25], Loureiro et al. [26], and Prado et al. [21], our results concerning genome size were similar to those for some of the cultivars measured in those works. Leal et al. [25] stated that the difference in DNA content between the analyses of Arumuganathan and Earle [23] and Lodhi and Reisch [24] and the ones they observed could be explained by the use of a plant standard instead of the chicken cells used in studies before 2000. Furthermore, the nuclear DNA amount data presented here were substantially larger than those of the *Vitis vinifera* accession in Chu et al. [27], 2C = 0.98 pg (with values for other *Vitis* species ranging from 0.94 to 1.24 pg). In this case, the differences cannot be attributed to the kind of internal standard used since, in both cases, the standard has plant origin. Additionally, in previous works in which different cultivars were compared, no relevant intraspecific variation was found, which points to the important stability of nuclear DNA content in this species, and in general in the genus *Vitis* and the whole family Vitaceae, since Arumuganathan and Earle [23], Lodhi and Reisch [24], and Chu et al. [27] studied other genera such as *Ampelopsis* and *Parthenocissus* as well as other species such as *V.* × *doaniana* Munson ex Viala, *V. rupestris* Scheele, and *V.* × *labruscana* L. H. Baley, with very uniform results as well. In agreement with the original idea of the constancy of genome size within a species, and irrespective of the fact that, as we stated, examples of intraspecific variation exist, *Vitis vinifera* joins other species in showing very small diversity in this parameter [29,30]. Our results differed from those Leal et al. [25], as statistically significant differences among landraces and between the same landraces were found, although the studied cultivars were not the same.

### 2.2. Genome Size Variation in Pityusic Vitis

As stated above, referring to the current dataset and the previous results available, *Vitis vinifera*, and the genus *Vitis* in general, showed a rather constant genome size. Nevertheless, some variation was detected and is worthy of mention.

Genome sizes estimated for each landrace without taking into account the island factor are summarised in Figure 1. As this figure shows, all means were distributed around 1.1 pg and 1.3 pg genome size values, but there were statistically significant differences between them (*F* = 2.563, *p* = 0.001).

Landraces ‘Santa Magdalena’, ‘Maçanet’, and ‘Fogoneu’ had significantly higher nuclear DNA content, while ‘Garnatxa’ and ‘Valencí tinto/Grumier’ cultivars had the lowest (Table 1).

When all means were compared, statistically significant differences were found between ‘AG2’ and ‘Santa Magdalena’ cultivars (*p* = 0.014); ‘Santa Magdalena’ and ‘Valencí tinto/Grumier‘ (*p* = 0.003) and ‘Garnatxa’ (*p* = 0.001); and between ‘Danugue’ and ’Garnatxa’ cultivars (*p* = 0.045); landrace ‘AG1’ and ‘Garnatxa’ also showed statistically significant differences (*p* = 0.046) (Appendix A).

Comparing the means of the Ibizan cultivars, the results showed statistically significant differences between landrace ‘AG2’ and ‘Santa Magdalena’ cultivar (*p* = 0.013); landrace ‘AG1’ and ‘Valencí tinto/Grumier’ (*p* = 0.044); ‘Santa Magdalena’ and ‘Valencí tinto/Grumier’ (*p* = 0.002); and between ‘Danugue’ and ‘Valencí tinto/Grumier’ (Appendix A). In Formentera cultivars, the results showed statistically significant differences between ‘Moscatell’ and ‘Garnatxa’ (*p* = 0.013) cultivars and between ‘Quigat’ and ‘Garnatxa’ cultivars (*p* = 0.048) (Appendix A).

A two-way ANOVA test demonstrated that there were no statistically significant differences between the same landraces cultivated in both islands.

Accessions grouped together as landrace ‘AG2’ and cultivar ‘Beba’ after the SSR markers results showed statistically significant differences within the same accession (F = 19.527 *p* = 0.000; *F* = 16.976 *p* = 0.026) (Table 2). None of the other grouped landraces or cultivars showed intraspecific differences.

Accessions grouped in ‘AG2’ had statistically significant differences between them. Significant differences were found between accession 5 and accessions 33 and 23, the latter being significantly different from all the accessions in the group. In ‘Beba’, accessions 1 and 26 showed statistically significant differences.

On-site observation for ‘AG2’ and ‘Santa Magdalena’ cultivars revealed that there were differences in grape colour, with ‘AG2’ having black grapes and ‘Santa Magdalena’ white grapes; the same was observed with ‘Santa Magdalena’ having significant differences with ‘Valencí tinto/Grumier’ and ‘Garnatxa’, which are both black grape varieties. Additionally, ‘Santa Magdalena’ was the first of all the accessions studied to ripen.

Concerning ‘AG2’ and ‘Beba’ cultivars, morphological differences were observed within the accessions in those groups, which could be associated with genome size variation (Figure 2).

In ‘AG2’, all the accessions had morphological differences. Accession 22 had a much shorter bunch than the other accessions in the same group and produced double bunches, while accession 33 had the longest bunch in this group. On the other hand, only accessions 5 and 33 had the same local name, ‘Monestrell’, while accessions 22 and 23 were called ‘Monestrell de xingló’ and ‘Monestrell d’Alger’ by farmers, respectively. Additionally, accession 23, which had a statistically significant different mean than the means of accessions 5, 22, and 23, from the AG2 group, was originally from Algeria, to which the past relatives of the farmer had often travelled (Marí, pers. comm. [31]). In the ‘Beba’ cultivar group, accessions 1 and 26 had differentially coloured grapes; nevertheless, the SSR results grouped them together [32].

Since no significant differences were found between the same cultivars or landraces in both islands selected for this study, the effects of insularity and those of the climatic, pedologic, and other ecological parameters should be discarded, as had occurred in other works such as those of Bennett et al. [33] or Chu et al. [27]. Irrespective of this fact, it may be considered that morphological differences, origin, or crop management can explain them. Hidalgo et al. [34] detected DNA content variation in plants grown in different conditions versus their wild relatives, and Stelzer et al. [35] found a positive correlation between genome size and body size, egg size, and embryonic development time in a population of *Brachionus asplanchnoidis*, although this could be due to the influence of other genes as well. If we take into account that smaller 2C content has been linked to environmental selection to thermal regulation [36] or to less time needed for seed ripening [37], this may help explain these results.

## 3. Materials and Methods

### 3.1. Plant Material and Origin

All samples considered in this study were cultivated in Ibiza and Formentera Islands, which, as already stated, are also known as Pytiusic Islands, belonging to the Balearic Islands, Spain.

The sampling comprised a total of 36 accessions, with different numbers of individuals investigated in each of the *Vitis vinifera* species (Table 1), 27 of which were from Ibiza and 9 were from Formentera; they encompassed 21 groups, 6 of which were local landraces from the Pityusic Islands [31], and the other ones were extended cultivars. Two of these landraces were named ‘AG1’ and ‘AG2’, as accessions with different local names were identified with SSR markers as the same cultivar and have yet to be assigned to one. Codes 1 to 28 corresponded to accessions from Ibiza Island, while code 4 was removed from this study because it had a doubtful origin and name. Accessions coded (29) to (37) were collected from Formentera Island. Origin, collectors, and dates are shown in Appendix A. Voucher specimens for each accession are deposited in the herbarium BC (Botanical Institute of Barcelona).

### 3.2. Genome Size Assessment

A small fragment of fresh young leaves of the studied plants was co-chopped using a razor blade with an internal standard in a ratio of 2:1 in 1200 μL of an LB01 buffer [38] (with 0.5% of Triton X-100 and supplemented with 100 µg/mL ribonuclease A (RNase A, Boehringer, Meylan, France) in a plastic Petri dish. *Petunia hybrida* Vilm. ‘PxPc6’ (2C = 2.85 pg; [39]) was used as an internal standard and was first separately analysed in 600 μL of the LB01 buffer to locate its peak position. Nuclei were filtered through a 70 μm nylon filter to eliminate cell debris before the addition of 36 μL of propidium iodide (1 mg/mL, solution in water; Invitrogen Eugene, Oregon, OR, USA). Samples were kept on ice before measurement. For each accession (Table 1), two samples of each individual were prepared and measured independently. Fluorescence analysis was carried out using an Epics XL flow cytometer (Coulter Corporation, Hialeah, Florida, FL, USA) at the Centres Científics i Tecnològics de la Universitat de Barcelona, with the standard configuration as described in Garnatje et al. [40]. The acquisition was stopped at 8000 nuclei. The DNA content was calculated for 10 of the aforementioned runs, assuming a linear correlation between the fluorescence signals (of the stained nuclei) and DNA amount. The mean and standard deviations were calculated for 2C values of each accession based on at least five individuals when possible. In the cases in which less than five individuals of each accession were measured, this is stated (Table 1). An example of fluorescence histogram is provided in Appendix A.

### 3.3. Statistical Analysis

The mean and the standard deviation (SD) of the GS and of the half-peak coefficient of variation (HPCV) were calculated for each accession using Microsoft Excel [41].

Regression analysis and a Kolmogorov–Smirnov test were carried out in order to test the normality of the data, as there were less than 50 observations. A test of analysis of variance (ANOVA) was performed to evaluate whether there were intra- and interspecific differences between the studied *V. vinifera* accessions from each island. When ANOVA resulted in a significant difference (*p* ≤ 0.05), a Student’s *t*-test was then conducted if there were only two accessions, and if there were more than two accessions, a post hoc Tukey test was used to compare the grouping of landraces by their means. The Wilcoxon test (non-parametric) was carried out to investigate whether there were differences among all the landraces considering only the island factor.

All tests were carried out with SPSS [42], and data visualisation was performed with the ggplot2 package [43] in R v.1.2.5042 [44].

## 4. Conclusions

The present results constitute the first dataset of the *Vitis* genome size in both studied islands, as well as, more broadly, in the Balearic Islands and Spain, and they are one of the largest prospections in this field for this species anywhere. We confirmed the constancy of the genome size in *Vitis vinifera* as, in general, in the genus, but some significant variations appeared, which were correlated with several morphological parameters such as bunch size or grape colour. More analyses should be conducted to explain the differences in nuclear DNA content found between landraces and cultivars and within them, as this could be due to different factors such as climatology, crop management, phenology, or colour mutations.

From the perspective of DNA content, it would be interesting to correlate landraces and cultivars with morphology and ripening data. Additionally, it could be studied whether crop management has any influence on genome size by comparing the same landraces and cultivars under the same conditions and comparing the landraces studied here with commercial cultivars and with landraces cultivated in other islands or countries to investigate whether genome size varies more or less in modern cultivars compared with local landraces and if there is any adaptation to diverse climatology conditions, crop management, and ripening characteristics.

## Figures and Tables

**Figure 1 plants-11-01892-f001:**
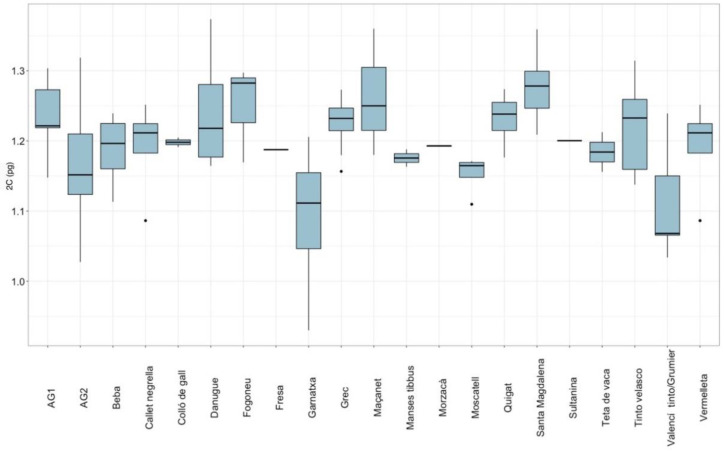
Grouped landraces boxplots for genome size (not considering island factor).

**Figure 2 plants-11-01892-f002:**
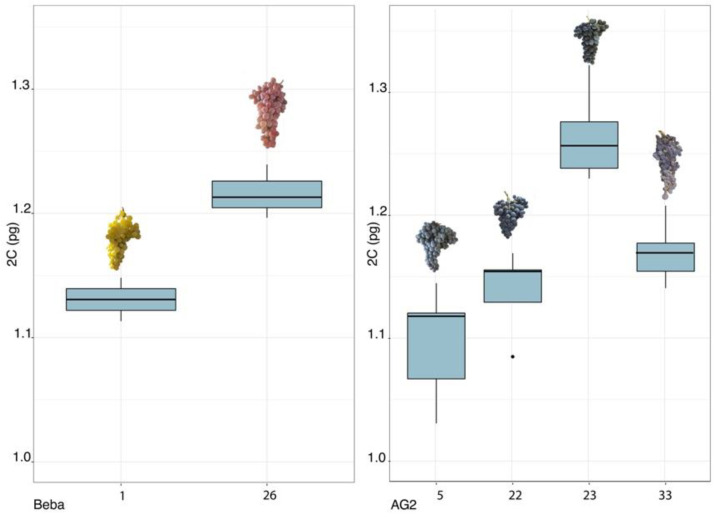
Boxplot with 2C values showing grouped accessions ‘Beba’ and ‘AG2’ with significant differences within them and indicating the shape and colour of the bunch in each case.

**Table 1 plants-11-01892-t001:** Nuclear DNA content of the studied *Vitis vinifera* cultivars and landraces.

Group	N ^1^	2C ± SD (pg) ^2^	1C (Mbp) ^3^	HPCV ± SD Plant	HPCV ± SD Standard
Santa Magdalena	7	1.277 ± 0.051	624.453	3.834 ± 1.441	1.909 ± 0.857
Maçanet	3	1.263 ± 0.091	617.607	3.242 ± 1.961	1.912 ± 1.400
Fogoneu	3	1.250 ± 0.070	611.250	3.180 ± 0.907	0.899 ± 0.431
Danugue	6	1.240 ± 0.081	606.360	4.008 ± 1.656	1.856 ± 0.901
Quigat	4	1.232 ± 0.041	602.448	3.794 ± 1.493	2.369 ± 0.902
AG1	9	1.229 ± 0.050	600.981	4.073 ± 1.091	2.129 ± 0.837
Grec	9	1.227 ± 0.039	603.426	4.473 ± 0.094	1.864 ± 0.880
Tinto velasco	5	1.221 ± 0.073	597.069	4.594 ± 1.881	1.569 ± 0.631
Sultanina	1	1.200 *	586.800	4.875 ± 0.233	0.955 ± 0.714
Colló de gall	2	1.198 ± 0.010	585.822	4.280 ± 0.421	1.103 ± 0.734
Morzacà	2	1.193 ± 0.001	583.377	3.990 ± 0.391	2.160 ± 0.944
Callet negrella	5	1.191 ± 0.064	582.399	4.592 ± 1.501	2.210 ± 1.374
Vermelleta	5	1.191 ± 0.064	582.399	4.366 ± 1.327	1.840 ± 1.174
Beba	7	1.188 ± 0.047	580.932	4.421 ± 0.939	1.961 ± 1.019
Fresa	1	1.188 *	580.932	4.845 ± 0.318	3.545 ± 0.403
Teta de vaca	4	1.184 ± 0.024	578.976	4.571 ± 1.346	3.068 ± 0.557
Manses tibbus	2	1.176 ± 0.018	575.064	3.043 ± 2.360	1.105 ± 0.527
AG2	20	1.164 ± 0.072	569.196	4.186 ± 1.669	1.583 ± 0.900
Moscatell	4	1.153 ± 0.029	563.817	3.705 ± 1.989	1.293 ± 0.863
Valencí tinto/Grumier	5	1.111 ± 0.083	543.279	4.347 ± 1.320	1.399 ± 0.795
Garnatxa	4	1.090 ± 0.117	533.010	4.888 ± 1.658	1.783 ± 0.659

^1^ Number of individuals measured; ^2^ 2C nuclear DNA content (mean value ± SD of individuals measured); Internal standard used *Petunia hybrida* Vilm (2C = 2.85 pg); ^3^ 1 pg = 978 Mbp [28]); * Only one individual was measured.

**Table 2 plants-11-01892-t002:** Significant differences within grouped accessions.

Landrace	Accession Code	Accession Code	*p* < 0.05
AG2	5	23	0
	5	33	0.024
	22	23	0
	23	33	0.004
Beba	1	26	0.026

## Data Availability

Raw data can be requested from the corresponding author.

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
