# Peer review of "Genome Size Variation Assessment in Vitis vinifera L. Landraces in Ibiza and Formentera (Balearic Islands)"

_plants, 2022, doi:10.3390/plants11141892_

Round 1
Reviewer 1 Report
The proposed manuscript “Genome size variation assessment in Vitis vinifera L. landraces in Ibiza and Formentera (Balearic Islands)” was re-submitted and the authors followed all my recommendations. Authors provided their internal R data and R data are consistent with proposed results from the manuscript. I did not find statistical tests for review but they seem to be trustworthy (performed in SPSS statistics software). I have not found any fundamental issues in the proposed ms. The ms is technically sound, presented in an intelligible fashion and written in good-quality English. Appropriately chosen terms and abbreviations are clearly explained and used for better understanding. Impact of the study is well highlighted.
After consideration of the manuscript quality I suggest accepting the manuscript for publication in the Plant journal.
Author Response
Thank you for your comments. We sincerely appreciate them.
Reviewer 2 Report
The report studied the genome size variations of 36 accessions including 15 cultivars and 6 landraces of Vitis vinifera L. of the Pityusic Islands in Spain. They were identified by the SSR markers and found to belong to 21 groups. The 2C values analyzed by the flow cytometry ranged from 1.09-1.28 pg. The accessions in landrace group ‘AG2’ and ‘Beba’ cultivars showed statistically significant differences. However, there were no statistically significant differences in the genome size content between Ibiza and Formentera islands. The current report only showed the genome size expected by the 2C values of the selected cultivars and landraces from Ibiza and Formentera islands. There is no important information available to confirm the connection of 36 accessions with known cultivars. In addition, the research flaws were found in the hypotheses and data analysis. It is not suggested for publication.
A. Page 2. It was stated that 39% of Balearic Islands vineyards are accounted for by ‘Cabernet’, ‘Sauvignon’, ‘Chardonnay blanc’, ‘Merlot’, and ‘Syrah’, 38% of the vineyards with local landraces such as ‘Manto negro’, ‘Callet’, and ‘Pensal Blanca’, and the remaining 23% consists of other varieties not specified. In addition, the main cultivars are ‘Monestrell’ and ‘Garrut’ with 76.45% of the crops In Ibiza and 57.5% are conformed of ‘Monestrell’, ‘Garrut’, and ‘Garnatxa tinta’ in Formentera. However, no known cultivars and landraces were analyzed for 36 accessions, even in the supplementary materials, except for accessions 5 and 33 were reported to have the same local name ‘Monestrell’, and accessions 22 and 23 were called ‘Monestrell de xingló’ and ‘Monestrell d’Alger’ on page 5.
B. The nuclear DNA content shown in Table 1 was not convincing. The whole-genome sequence of Vitis vinifera L. was known to be ~500 Mb. The data presented in this study were all larger than expected. Though the authors explained it in Section 2.1, however, it needs other experimental data to prove the speculations.
C. Table 1. The use of taxon for each cultivar or landrace was inappropriate. A taxon is usually defined as a particular name. Using the “group”, as stated in the text, would be better. Moreover, the HPCV for the standards was variable but the standards were all the same, Petunia hybrida Vilm. ‘PxPc6’. It needs further explanation. In addition, the “Grec” was not found in the supplementary materials among all the groups but the “Llora”.
D. The association of the shape and color of the bunch to the genome size variations was only speculation. It needs further evidence.
E. Figure S1. The fluorescence histogram of accession ‘Moscatell’ (3) needs to be further explained. How to calculate the genome size using Petunia hybrida as the standard? In addition, counts are usually used as the unit of the y-axis.
F. Figures 1 and 2. The use of GS as the unit of the y-axis was not appropriate. Using Genome Size - 1C (pg) would be better.
G. The P-values for the comparison of the Ibizan cultivars were all wrong according to Table S2.
H. The P-value for the accessions grouped in ‘AG2’ between accessions 33 and 22 was missing in Table 2.
I. The 2C values of ‘Santa Magdalena’ was 1.277 pg according to the data provided in Table 1.
J. The vineyard surface areas in Spain were not updated.
K. The ideas of detection and evaluation of intraspecific variations and their links to the distribution were not clarified.
L. Section 3 was missing.
Round 2
Reviewer 2 Report
The report studied the genome size variations of 36 accessions including 15 cultivars and 6 landraces of Vitis vinifera L. of the Pityusic Islands in Spain. They were identified by the SSR markers and found to belong to 21 groups. The 2C values analyzed by the flow cytometry ranged from 1.09-1.277 pg. The accessions in landrace group ‘AG2’ and ‘Beba’ cultivars showed statistically significant differences. However, there were no statistically significant differences in the genome size content between Ibiza and Formentera islands. Though the current report only showed the genome size expected by the 2C values of the selected cultivars and landraces from Ibiza and Formentera islands by the flow cytometric method, the manuscript has been revised to a relatively acceptable edition. To reflect the truth, though some issues still remain, however, it is suggested for publication in the journal Plants after minor modifications. The suggestions for the improvement of the manuscript are listed below.
A. Due to there is still a 23% percentage of cultivars unknown in those vineyards, only accessions 5 and 33 were reported to have the same local name ‘Monestrell’, and accessions 22 and 23 were called ‘Monestrell de xingló’ and ‘Monestrell d’Alger’ on page 5 in this study. The facts should be claimed that the other cultivars and landraces were unknown.
B. Though nuclear DNA assessment by flow cytometry showed that the genome sizes were estimated to be between 533.01-624.453 Mbp, the fact that the whole-genome sequence of Vitis vinifera L. was known to be ~500 Mb still needs to be mentioned. The approaches for the confirmation of actual genome sizes could be described to prove the speculations from the results of flow cytometry.
C. Though further evidence cannot be made until more studies have been made, which we remark in section 4 Conclusions, the methods for the study of the association of the shape and color of the bunch to the genome size variations could be described.
D. Figure S1. Due to the genome size of Petunia hybrida being known, the genome size should be described in the figure legend because it was used as the standard. The calculation of the genome size of accession‘Moscatell’(3) has to be described in the legend too because it was the only example shown in this study. In addition, counts are usually used as the unit of the y-axis to replace the use of events.
E. The P-values for the comparison of the Ibizan cultivars, described on page 5, between lines 141-144, were all wrong according to Table S2. For example, the P-value between landrace ‘AG2’ and ‘Santa Magdalena’ was 0.013 but not 0.016, between landrace ‘AG1’ and 142 ‘Valencí tinto’ was 0.044 but not 0.013, between ‘Santa Magdalena’ and ‘Valencí tinto/Grumier’ was 0.002 but not 0.003, and between ‘Danugue’ and ‘Valencí tinto/grumier’ was 0.007 but not 0.003.
F. The 2C value of ‘Santa Magdalena’ was 1.277 pg according to the data provided in Table 1. The value, 1.28, of which was different in the abstract and on page 2, line 94. The value has to be accurate.
G. Though it was stated that the “two-way ANOVA test demonstrates that there are no statistically significant differences between the same landraces cultivated in both islands”, the fact that it may be to the cultivars and landraces selected in this study should be mentioned. The presence of intraspecific variations and their links to the distribution could not be excluded. A future investigation may also need to be performed to clarify the issue.
H. The method for the identification of cultivars and landraces by the SSR markers should be described.
I. Table 1. The last column “Standard” was the same in each case and it was already mentioned as a note in this table. It is suggested to delete this column. In addition, some of the values shown in Table 1 were calculated to two decimal places but some were calculated to three decimal places. It is suggested to calculate to three decimal places for all the values.
Author Response
Please see the attachment.

This manuscript is a resubmission of an earlier submission. The following is a list of the peer review reports and author responses from that submission.
Round 1
Reviewer 1 Report
The manuscript described the first dataset of Vitis genome size of registered cultivars and landraces in the Balearic Islands and Spain using flow cytometry with propidium iodide. The authors compared the optained result with previous results of Vitis genome size. Statistically significant differences in genome size were detected between samples. The current study indicates that more research should be provided to understand the changes in nuclear DNA content reported between landraces and cultivars.
I recommended the article for publication because the manuscript provided new knowledge about Vitis genome size of cultivars and landraces.
Reviewer 2 Report
The proposed manuscript “Genome size variation assessment in Vitis vinifera L. landraces in Ibiza and Formentera (Balearic Islands)” contributes to the study of genome size between different populations and landraces of Vitis vinifera. Study showed no significant variation in genome size between populations from different islands. Significant difference was found in genome size between some kultivars. I have not found any fundamental issues in the proposed ms. The ms is technically sound, presented in an intelligible fashion and written in good-quality English. Appropriately chosen terms and abbreviations are clearly explained and used for better understanding. Impact of the study is well highlighted. I have some minor suggestions:
- The most important one is that it is not clear to me which cultivars/landraces were included and originated from different islands. Did you compare genome sizes of individuals of the same landraces (considering island factor)? Could you highlight this information in the manuscript?
- Put author’s addresses consistently (by same way) (lines 5-9).
- “Significant differences in between them”. There is probably typo “in” and it should be deleted (line 134).
- Table 3 is too long. Information here is quite important because there is an origin of each cultivar. This information can be summarized as a statistical description of variability on genome size. I suggest putting this table as the supplementary material.
- “We confirm the constancy of this parameter”. I am not sure what parameter you mean.
- Statistical analysis should be available for readers. Authors correctly stated that raw data can be requested. For the second round of the review, please upload raw data including R scripts with your first revised version.
After consideration of the manuscript quality I suggest a major revision and to follow all the suggestions above.
Reviewer 3 Report
Thank you for the opportunity to review the MS, ID Plants-1708251.
The methodology and approach for the aim of the MS are effectively conducted. The authors have measured C2 of 36 accessions of the Vitis vinifera species and found consistency in their DNA content. However, some variation was observed and claimed to possibly correlate with several morphological properties such as bunch size or grape colours.
The reported finding here, although useful, is not adequate for a scientific paper to be published. It lacks novelty and substantial content. I would recommend the authors to do extra work to add to this paper in the future such as:
- Genome sequencing the accessions, if not all then some representatives which show differences in this study
- Carrying out genomic comparative study of the sequenced accessions
- Conduct genotyping and phenotyping at least 150-200 more accessions worldwide to see any association between morphology and genotypes
- The genotype data set obtained can be used for genetic diversity analysis in the future.
Thank you and kind regards
Reviewer 4 Report
The study investigated the genome size of fifteen cultivars and six landraces of Vitis vinifera L. which were identified by the SSR markers. A total of thirty-one accessions, from Ibiza and Formentera in Spain, were analyzed by the flow cytometric analysis. The results indicated that the 2C values of various grapevine cultivars and landraces ranged from 1.09-1.28 pg which showed statistically significant differences. Nevertheless, the authors reported that the same cultivars and landraces showed no statistically significant differences between the two islands. But statistically significant differences were found between the accessions of Beba cultivar and AG2 landrace. Though the study was locally performed on grapevine grown in Ibiza and Formentera in Spain, it may establish a basis for the comparison of grapevine genome variations between different areas by an efficient approach without genome resequencing. However, the report in its current form does not meet the standards for publication. One is due to the loss of essential data and information in the manuscript, and the other one is the lack of importance of the contribution of this study. Therefore, the manuscript is not recommended for publication in the journal Plants. The major concerns, for example, are listed below.
- Though the information on grapevine cultivars and landraces were shown in Table 3, however, the details of voucher specimens should be described, including collection numbers and/or specimen numbers. Only code was shown for each cultivar or landrace. It may be hard for other researchers to perform other studies if the voucher information is not available. In addition, code (4) was missing in this study without explanation. Furthermore, code (27) was not Beba cultivar. It raised the possibility that the result was doubted on the significant differences within grouped Beba accessions shown in Table 2 and the discussion on page 5, though the data shown in Figure 2 may be correct.
- The morphological and phenotypical information of various grapevine cultivars and landraces were not provided. For example, Sultanina is one of the most important tables grape varieties for providing the seedlessness phenotype.
- The data of the fluorescence histogram of the genome size analysis was not provided.
- The information on grapevine cultivation discussed in the introduction maybe not be correct. According to the reports of The European Cooperative Programme for Plant Genetic Resources (ECPGR) and The International Organisation of Vine and Wine (OIV), accessed on April 20, 2022, the area of production is over 7,464 Mha across the world. In addition, the grapevine varieties are over 8,400 in the world.
- The purposes of this study, including the detection and evaluation of intraspecific variations and their links to the distribution, were not clearly discussed.
- The definition of geographical locations was confounded. The Ibiza and Formentera islands, Balearic Islands, and Pityusic islands were irregularly used. The fact should be reflected that only the grapevine varieties cultivated on Ibiza and Formentera islands were studied.